# The Lattice Geometry of Neural Network Quantization: A Short Equivalence Proof of GPTQ and Babai's Algorithm

**Johann Birnick**
Department of Mathematics
University of California San Diego
San Diego, CA 92093, USA
jbirnick@ucsd.edu

## Abstract

We explain how data-driven quantization of a linear unit in a neural network corresponds to solving the closest vector problem for a certain lattice generated by input data. We prove that the GPTQ algorithm (Frantar et al., 2023) is equivalent to Babai's well-known nearest-plane algorithm (Babai, 1986). We furthermore provide geometric intuition for both algorithms. Lastly, we note the consequences of these results, in particular hinting at the possibility of using lattice basis reduction for improved quantization.

## 1 Quantization and Lattices

Computations in neural networks are usually carried out in 32-bit or 16-bit floating point arithmetic. In particular, the parameters (weights) of the network are stored in this comparatively high precision.

*Quantization* is the art of reducing precision, in favor of less memory consumption and faster computation, while keeping the accuracy as high as possible. In this paper, we are interested only in *post-training quantization of the weights*: We are handed a trained neural network, and our goal is to approximate (some of) the parameters of the network with a coarse numerical alphabet, while keeping the accuracy high.

Commonly, this effort is focused on the *linear* parts of the network. That is, we are given a linear map $\mathbb{R}^n \to \mathbb{R}^m$, represented by a weight matrix $W \in \mathbb{R}^{m \times n}$, and we seek to find another $m \times n$ matrix $V$, whose entries have lower numerical precision and which "approximates $W$ well". Concretely, this means:

**Low numerical precision.** We will model $V$ as having *integer* entries, i.e., $V \in \mathbb{Z}^{m \times n}$. Together with a simple scaling, this covers the case when the available low-precision alphabet is $\alpha\mathbb{Z}$ for some $\alpha \in \mathbb{R}$. Up to clipping, this models quantization with low-bit integers as a memory/computation unit.

**Approximation of $W$.** The approximation problem is carried out in a *data-driven* context: We are allowed to sample representative inputs $x_1, x_2, ..., x_k \in \mathbb{R}^n$ of the linear unit $W$. Indeed, in practice we can just take some of the training inputs and send them through the network until they reach the linear unit of interest. We want $V$ to approximate $W$ well *on these specific inputs*, so we want to minimize:

$$\sum_{j=1}^{k} \|Wx_j - Vx_j\|_2^2 = \sum_{j=1}^{k}\sum_{i=1}^{m} \langle W_{i,:} - V_{i,:}, x_j \rangle^2 = \sum_{i=1}^{m} \|XW_{i,:}^T - XV_{i,:}^T\|_2^2 \tag{1}$$

Here, $X \in \mathbb{R}^{k \times n}$ is the matrix with rows $x_1, ..., x_k$, and $W_{i,:}$ is the $i^{\text{th}}$ row of $W$. Note that this optimization problem is *separable*: to minimize the sum on the right, it suffices to minimize each summand $\|XW_{i,:}^T - XV_{i,:}^T\|_2^2$ separately. This corresponds to quantizing a single neuron at a time. So the problem is:

**Problem.** Given $X \in \mathbb{R}^{k \times n}$ and $w \in \mathbb{R}^n$, find $v \in \mathbb{Z}^n$ so that $\|Xw - Xv\|_2$ is as small as possible.

**Lattice view.** We will now explain the connection to lattices. A *lattice* is the $\mathbb{Z}$-span $\mathbb{Z}b_1 + ... + \mathbb{Z}b_n$ of a set of $\mathbb{R}$-linearly independent vectors $b_1, ..., b_n$ in $\mathbb{R}^k$. The vectors $b_1, ..., b_n$ are called a *basis* of the lattice. Multiple different bases can produce the same lattice. For example, $\mathbb{Z}^2 \subset \mathbb{R}^2$ is a lattice, and two possible bases are $\{(1, 0), (0, 1)\}$ and $\{(3, 1), (5, 2)\}$. See, for instance, Micciancio & Goldwasser (2002) for more background on lattices.

In the problem above, we can view the columns of $X$ as the basis for a lattice in $\mathbb{R}^k$. That is assuming these columns are linearly independent; more on that below. Then $Xw$ can be just viewed as a point in $\mathbb{R}^k$, and $Xv$ is a lattice point (an element of the lattice). As $v$ runs through $\mathbb{Z}^n$, $Xv$ runs through all the lattice points. So the minimization problem above asks to compute a lattice point which is close to $Xw$. In the lattice community this is known as the (approximate) *closest vector problem* (CVP).

While this is generally NP-hard to solve optimally, decades of research have been devoted to practical algorithms that approximately solve CVP. The common approach is to employ an LLL-like algorithm for *basis reduction* (Nguyen & Vallée, 2010), followed by Babai's nearest plane algorithm (Babai, 1986). We will see in section 2 that GPTQ (Frantar et al., 2023) is exactly equivalent to Babai's algorithm (up to possibly reversing the columns of $X$).

**Regularization.** Note that the columns of $X$ might not be linearly independent; in particular this will be the case if the number $k$ of calibration inputs is less than the number of features $n$. However, we can use the following regularization: Append a scalar multiple of the $n \times n$ identity matrix below $X$, so that

$$X' := \begin{pmatrix} X \\ \mu \cdot I_{n \times n} \end{pmatrix} \qquad \text{where } \mu > 0 \tag{2}$$

is used in place of $X$. The columns of $X'$ are linearly independent, and choosing $\mu \to \infty$ will lead to the naive quantization $v := \text{round}(w)$ as the optimal solution.

When $\mu = \sqrt{\lambda}$, this is equivalent to the $\lambda$-regularization in Frantar et al. (2023). Indeed, GPTQ works with the matrix $X^T X$, and for regularization it replaces this matrix with $X^T X + \lambda I$ instead. But we have:

$$X'^T X' = X^T X + (\mu I)^T (\mu I) = X^T X + \lambda I \quad , \tag{3}$$

So the $X'$ regularization will yield the same result, but it also admits a lattice interpretation.

In summary, we showed how quantization of linear units reduces to solving the CVP for a lattice generated by input data. One can now apply the full range of CVP algorithms for neural network quantization, provided one can scale them to the large lattices that are involved in quantization, see section 3.

**Concurrent Work.** A related paper by Chen et al. (2026) is being published at the same time as this work and shows very similar results to those we present here. We want to emphasize that our work was conducted fully independently and had been in development for an extended period of time. Our proofs follow a different approach than Chen et al. (2026) and are shorter; we believe that they offer a concise and conceptually elegant perspective.

## 2 GPTQ IS EQUIVALENT TO BABAI'S ALGORITHM

In this section, we view the GPTQ algorithm (Frantar et al., 2023) and Babai's nearest plane algorithm (Babai, 1986) as procedures for solving the problem from section 1:

**Problem.** Given $X \in \mathbb{R}^{k \times n}$ and $w \in \mathbb{R}^n$, find $v \in \mathbb{Z}^n$ so that $\|Xw - Xv\|_2$ is as small as possible.

We will show that the algorithms are equivalent, up to reversing the basis of the lattice. We will first review both GPTQ and Babai's algorithm separately. We will see that they differ in two aspects:

- GPTQ works in "parameter space" $\mathbb{R}^n$. Babai's algorithm works in "data space" $\mathbb{R}^k$.

- As GPTQ progresses to smaller sublattices, it keeps the target contained in the $\mathbb{R}$-span of the sublattice. (In every iteration it projects onto this span.) BABAI omits such a projection as it's not necessary.

In a nutshell, the two algorithms are related by a composition of linear projections $\mathbb{R}^k \to \mathbb{R}^n \to \mathbb{R}^{n-i}$. The core is the map $\mathbb{R}^k \to \mathbb{R}^n$, which is given by $X^+$ and projects the "data space" onto the "parameter space". The map $\mathbb{R}^n \to \mathbb{R}^{n-i}$ is the projection onto the last $n - i$ coordinates, in step $i$ of the algorithm, and corresponds to projecting onto the remaining sublattice. Whatever Babai's algorithm does in $\mathbb{R}^k$, project it down to $\mathbb{R}^{n-i}$, and this yields precisely what GPTQ does.

The formal equivalence proof then proceeds by essentially rewriting both algorithms as recursive algorithms and using the insight above.

**Notation.** We will use the following notation. When $A$ is a matrix, then $A_j$ denotes its $j^{\text{th}}$ column, and $A_{i,:}$ denotes its $i^{\text{th}}$ row. With $A_{\geq j}$ we denote the submatrix of $A$ which omits the first $j - 1$ columns, and $A_{\geq i, \geq j}$ denotes the submatrix which omits the first $i - 1$ rows and $j - 1$ columns. For vectors $a$, we denote by $a_i$ the $i^{\text{th}}$ coordinate, and $a_{\geq i}$ denotes the subvector which omits the first $i - 1$ coordinates.

$X$ and $w$ are fixed throughout. We have already seen how one can apply regularization to $X$, so from now on we will assume that $X$ has linearly independent columns.

## 2.1 GPTQ ALGORITHM REVISITED

The original description of GPTQ (Frantar et al., 2023) computes a matrix $\tilde{L}$ as the Cholesky decomposition of $(X^T X)^{-1}$:

$$\tilde{L}\tilde{L}^T = (X^T X)^{-1} \tag{4}$$

We will now show that this just corresponds to taking a QL-decomposition of $X$ and inverting $L$. Suppose

$$X = QL \tag{5}$$

with $Q \in \mathbb{R}^{k \times n}$ having orthonormal columns and $L \in \mathbb{R}^{n \times n}$ being lower triangular with positive entries on the diagonal. Then we have:

$$\tilde{L}\tilde{L}^T = (X^T X)^{-1} = (L^T Q^T Q L)^{-1} = (L^T L)^{-1} = L^{-1} L^{-T} \tag{6}$$

From the uniqueness of the Cholesky decomposition ($X^T X$ is positive definite) we get:

$$\tilde{L} = L^{-1} \tag{7}$$

So in GPTQ we can also compute $\tilde{L}$ without a Cholesky decomposition; instead we compute the QL-decomposition of $X$ and then invert $L$. However, this is mostly useful for theoretical study. In practice, one usually uses a lot of calibration data, $k \gg n$, in which case it's more memory-efficient to only accumulate the Gram matrix $X^T X$ instead of storing the full matrix $X$.

With this note about $\tilde{L}$ in place, the GPTQ algorithm can be described as follows: (see section A)

```
 1: procedure GPTQ(X, w)
 2:     Compute QL = X.                          ▷ QL-decomposition with L_{i,i} > 0
 3:     L̃ ← L^{-1}
 4:     w^{(0)} ← w
 5:     for i = 1, ..., n do
 6:         v_i ← round(w_i^{(i-1)})
 7:         Δ_i ← v_i - w_i^{(i-1)}
 8:         w^{(i)} ← w^{(i-1)} + (Δ_i / L̃_{i,i}) · L̃_i
 9:     end for
10:     return v
11: end procedure
```

The idea behind this is to find a $w'$ whose first coordinate $w_1'$ is fixed to be $\text{round}(w_1)$, and which minimizes $\|Xw - Xw'\|$. This can then be applied recursively to the other coordinates. The optimization problem to find this $w'$ has the explicit solution

$$w' = w + \frac{\text{round}(w_1) - w_1}{\tilde{L}_{1,1}} \cdot \tilde{L}_1 \qquad (8)$$

which then yields the GPTQ procedure described above. See Hassibi et al. (1993), Frantar & Alistarh (2022), and Frantar et al. (2023) for the history and derivation.

Note that, since $\tilde{L}_i$ has the first $i-1$ coordinates all set to zero, the coordinates of the $w^{(i)}$ stabilize as $i$ increases. Also, because of the normalization factor $\Delta_i / \tilde{L}_{i,i}$ and the definition of $\Delta_i$, the coordinates that they stabilize to are the coordinates of $v$. Concretely, we have $w_i^{(j)} = v_i$ for $i \leq j$. In particular, $w^{(n)} = v$.

As already noted, one might as well write GPTQ as a recursive algorithm:

1: **procedure** GPTQ-REC$(X, w)$
2:     Compute $QL = X$.                                    ▷ QL-decomposition with $L_{i,i} > 0$
3:     $\tilde{L} \leftarrow L^{-1}$
4:     $v_1 \leftarrow \text{round}(w_1)$
5:     $v_{\geq 2} \leftarrow$ GPTQ-REC$(X_{\geq 2}, (w + \frac{v_1 - w_1}{\tilde{L}_{1,1}} \tilde{L}_1)_{\geq 2})$
6:     **return** $v$
7: **end procedure**

Indeed, the equivalence of the two procedures follows from observing that the QL-decomposition of $X_{\geq 2}$ is precisely given by $Q_{\geq 2}$ and $L_{\geq 2, \geq 2}$, and that the inverse of $L_{\geq 2, \geq 2}$ is given by $\tilde{L}_{\geq 2, \geq 2}$:

$$X_{\geq 2} = Q_{\geq 2} \cdot L_{\geq 2, \geq 2} \qquad (L_{\geq 2, \geq 2})^{-1} = (L^{-1})_{\geq 2, \geq 2} = \tilde{L}_{\geq 2, \geq 2} \qquad (9)$$

From these equalities one can see that GPTQ-REC is equivalent to GPTQ. (Formally one could prove it via induction.)

## 2.2 BABAI'S ALGORITHM

We will now describe Babai's nearest plane algorithm (Babai, 1986), which was developed in the context of lattices. Recall that we view the columns of $X$ as the basis for an $n$-dimensional lattice in $\mathbb{R}^k$. We want to find a lattice vector close to $Xw$. The idea is to maintain a target vector $t$, initialized with $t = Xw$. Then one builds up $v$ by taking inner products of $t$ with the Gram-Schmidt basis vectors associated to the lattice basis. This can be interpreted as finding a certain "nearest plane", hence the name; see section 2.3 and Nguyen & Vallée (2010, Chapter 6).

The normalized Gram-Schmidt basis can be seen as the $Q$-factor in a QR-decomposition of $X$. The lengths of the Gram-Schmidt basis vectors are stored in the diagonal elements of the $R$-factor. We will instead use the QL-decomposition here, so that it is compatible with GPTQ; this simply corresponds to applying the "usual" Babai algorithm (which uses a QR-decomposition) on the reversed lattice basis. (See section A for details of the relation to the classic algorithm.)

1: **procedure** BABAI$(X, w)$
2:     Compute $QL = X$.                                      ▷ QL-decomposition with $L_{i,i} > 0$
3:     $t^{(0)} \leftarrow Xw$
4:     **for** $i = 1, ..., n$ **do**
5:         $v_i \leftarrow \text{round}(\frac{\langle t^{(i-1)}, Q_i \rangle}{L_{i,i}})$
6:         $t^{(i)} \leftarrow t^{(i-1)} - v_i X_i$
7:     **end for**
8:     **return** $v$
9: **end procedure**

BABAI could also be written as a recursive algorithm, although then it should not take $w$ as input but rather directly the target vector $t = Xw$, which for the recursion would be replaced by $t - v_1 X_1$. We omit the recursive version of the procedure here; instead it will be implicit in the equivalence proof in section 2.4.

*Remark.* We note that BABAI only depends on the intrinsic geometry of the lattice, namely the Gram matrix $X^T X$. It is therefore invariant under the transformation $X \mapsto UX$ for any $U \in \mathbb{R}^{l \times k}$ for which $U^T U$ acts as the identity on the column space of $X$, as this transformation preserves the Gram matrix $X^T X$. Thus, in practice, one would replace $X$ by $L$, where $QL = X$ is the QL-decomposition of $X$. This is much more efficient for memory and computation, because $X \in \mathbb{R}^{k \times n}$ whereas $L \in \mathbb{R}^{n \times n}$, and $k \gg n$, see section B in the appendix. For this paper, however, we chose to use $X$ everywhere to highlight the distinct spaces in which computations are carried out.

## 2.3 THE UNDERLYING GEOMETRY

We will now provide geometric intuition for what GPTQ and Babai's algorithm are doing. Note that there are fundamentally two spaces at play: The "parameter space" $\mathbb{R}^n$, in which $w$ (and $v$) lives, and the "data space" $\mathbb{R}^k$ in which $Xw$ (and $Xv$) lives. The matrix $X$ can be seen as an embedding map $\mathbb{R}^n \hookrightarrow \mathbb{R}^k$, which maps the quantization grid $\mathbb{Z}^n$ to a lattice in $\mathbb{R}^k$.

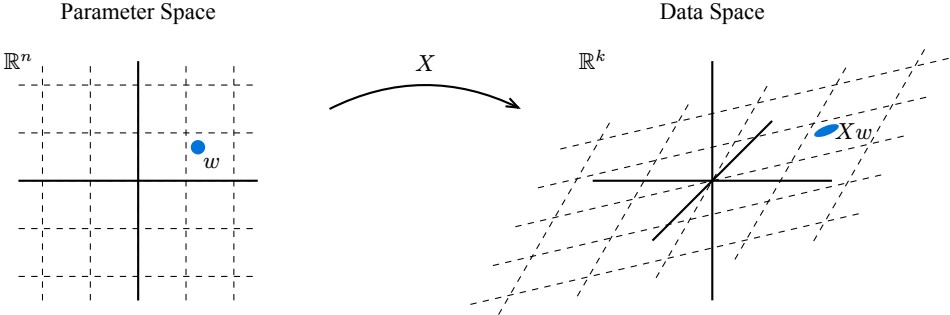

Figure 1: Two spaces at play. $X$ embeds $\mathbb{R}^n$ into $\mathbb{R}^k$, mapping the quantization grid $\mathbb{Z}^n$ to a lattice in $\mathbb{R}^k$. GPTQ works in $\mathbb{R}^n$, on the left. Babai's algorithm works in $\mathbb{R}^k$, on the right.

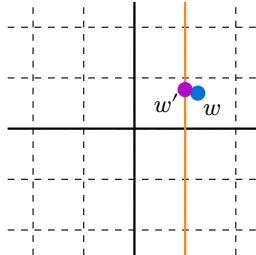

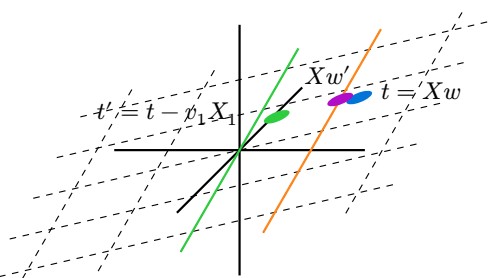

Figure 2: GPTQ fixes $v_1 := \mathrm{round}(w_1)$. This restricts $v$ to lie on the orange plane. It defines a new target weight $w'$ on the orange line, and then proceeds recursively. The target weight $w'$ is *not* just an orthogonal projection to the orange plane. Instead, the update step implicitly uses the geometry from the lattice; in $\mathbb{R}^k$ on the right it indeed corresponds to an orthogonal projection.

Figure 3: Babai's algorithm looks for the nearest plane (parallel to the orange or green plane) to $t$. It identifies the orange plane, and then subtracts an appropriate integer multiple of $X_1$ from $t$, leading to the green point $t'$. It then recursively looks for a lattice point in the green sublattice which is close to the green point $t'$.

Figures 2 and 3 demonstrate what happens in the first step of each algorithm. As we will prove below, both GPTQ and BABAI compute the same first coordinate of $v$, i.e., $v_1$, but they do it in different ways.

Note that BABAI's new target vector $t'$ does *not* lie in the $\mathbb{R}$-span of the green sublattice. (In fact, at the very end we will have $t = Xw - Xv$.) One could project it onto that span, and the result for $v$ wouldn't change. Indeed, the difference coming from the projection is, by definition, orthogonal to the green plane, so it doesn't matter in future computations. We will use this in the equivalence proof below, because GPTQ always does this projection implicitly.

## 2.4 PROOF OF EQUIVALENCE

**Theorem 2.1.** *The procedures* GPTQ *and* BABAI *are equivalent. That is, for any* $X \in \mathbb{R}^{k \times n}$ *(of full column rank) and* $w \in \mathbb{R}^n$, *they produce the same output* $v \in \mathbb{Z}^n$.

*Proof.* We already saw that GPTQ is equivalent to GPTQ-REC. We will now show that both GPTQ-REC and BABAI are equivalent to the following procedure, which can be interpreted as a recursive version of BABAI with an additional projection step to the remaining sublattice as noted in section 2.3.

1: **procedure** BABAI-PROJ-REC$(X, w)$
2:     Compute $QL = X$.                                    ▷ QL-decomposition with $L_{i,i} > 0$
3:     $\tilde{L} \leftarrow L^{-1}$
4:     $t \leftarrow Xw$
5:     $v_1 \leftarrow \text{round}(\frac{\langle t, Q_1 \rangle}{L_{1,1}})$
6:     $v_{\geq 2} \leftarrow$ BABAI-PROJ-REC$(X_{\geq 2}, (w + \frac{v_1 - w_1}{\tilde{L}_{1,1}} \tilde{L}_1)_{\geq 2})$
7:     **return** $v$
8: **end procedure**

GPTQ-REC *is equivalent to* BABAI-PROJ-REC. The only difference between the two algorithms is how they compute $v_1$. GPTQ just rounds $w_1$, and BABAI-PROJ-REC rounds the following quantity:

$$\frac{\langle t, Q_1 \rangle}{L_{1,1}} = \frac{\langle Xw, Q_1 \rangle}{L_{1,1}} = \frac{Q_1^T QLw}{L_{1,1}} = \frac{e_1^T Lw}{L_{1,1}} = \frac{L_{1,:}w}{L_{1,1}} = w_1 \tag{10}$$

BABAI *is equivalent to* BABAI-PROJ-REC. This is less obvious. Consider the value of "$t$" in the first BABAI-PROJ-REC recursion, i.e. in the first nested call caused by line 6. It is equal to:

$$X_{\geq 2} \left( w + \frac{v_1 - w_1}{\tilde{L}_{1,1}} \tilde{L}_1 \right)_{\geq 2} \tag{11}$$

Suppose that this product would equal $t - v_1 X_1$. Then it would follow by induction that BABAI-PROJ-REC is equivalent to BABAI.

We will show that the product *almost* equals $t - v_1 X_1 = Xw - v_1 X_1$, up to an additive term of $\kappa Q_1$, where $\kappa \in \mathbb{R}$ is some scalar. And indeed this suffices to show the equivalence:

Note that, in future iterations/recursions, both BABAI and BABAI-PROJ-REC will only ever use $t^{(i)}$ (or "$t$") to take inner products with $Q_2, ..., Q_n$, and these vectors are orthogonal to $Q_1$. So one could, for example, modify BABAI so that $t^{(i)}$ gets additively shifted by $\kappa_i Q_i$ for any $\kappa_i \in \mathbb{R}$, and the output wouldn't change. Concretely, let's define:

1: **procedure** BABAI$_{\kappa_1, ..., \kappa_n}(X, w)$
2:     Compute $QL = X$.                                    ▷ QL-decomposition with $L_{i,i} > 0$
3:     $t^{(0)} \leftarrow Xw$
4:     **for** $i = 1, ..., n$ **do**
5:         $v_i \leftarrow \text{round}(\frac{\langle t^{(i-1)}, Q_i \rangle}{L_{i,i}})$
6:         $t^{(i)} \leftarrow t^{(i-1)} - v_i X_i + \kappa_i Q_i$
7:     **end for**
8:     **return** $v$
9: **end procedure**

The previous paragraph shows that BABAI is equivalent to BABAI$_{\kappa_1, ..., \kappa_n}$ for *any* $\kappa_1, ..., \kappa_n \in \mathbb{R}$. And the equivalence of BABAI-PROJ-REC with BABAI$_{\kappa_1, ..., \kappa_n}$ for some *specific* $\kappa_1, ..., \kappa_n \in \mathbb{R}$, which depend on the input $(X, w)$, follows from an induction argument, given the claim above.

It remains to prove the claim:

$$X_{\geq 2}\left(w + \frac{v_1 - w_1}{\tilde{L}_{1,1}}\tilde{L}_1\right)_{\geq 2} = X\left(w + \frac{v_1 - w_1}{\tilde{L}_{1,1}}\tilde{L}_1\right) - X_1\overbrace{\left(w + \frac{v_1 - w_1}{\tilde{L}_{1,1}}\tilde{L}_1\right)_1}^{=v_1} \quad (12)$$

$$= X\left(w + \frac{v_1 - w_1}{\tilde{L}_{1,1}}\tilde{L}_1\right) - v_1 X_1 \quad (13)$$

$$= Xw - v_1 X_1 + \underbrace{\frac{v_1 - w_1}{\tilde{L}_{1,1}}}_{\substack{\text{the previously} \\ \text{mentioned } \kappa}} \underbrace{Q_1}_{=X\tilde{L}_1} \quad (14)$$

$\square$

## 3 CONSEQUENCES AND FUTURE WORK

The equivalence of Babai's algorithm with GPTQ has direct consequences for quantization with GPTQ.

**Correct Handling of Quantization Over Multiple Layers.** Suppose we have quantized a linear unit of a neural network, and then we want to quantize another linear unit which comes later in the network. Recall that in order to obtain $X$, we take sample inputs of the neural network, and send them through the network to just before our linear unit. This means the data will pass units that we have already quantized. Usually we want to pass the data through the *quantized* units to generate the lattice $\hat{X}$, while passing it through the *original* units to generate the target $Xw$. So we now want to find $v \in \mathbb{Z}^n$ which minimizes:

$$\|Xw - \hat{X}v\|$$

While for GPTQ it's not obvious how to deal with this modified problem, it's completely obvious for Babai's algorithm. Indeed, we just need to set the target vector to $t = Xw$. (And work with the lattice generated by $\hat{X}$.) If one wants to use GPTQ instead, that corresponds to projecting $Xw$ down onto the $\mathbb{R}$-span of the lattice $\hat{X}$; the projected point will be equal to $\hat{X}\hat{w}$ for some $\hat{w} \in \mathbb{R}^n$, which should then be used as an input to GPTQ. I.e., one should run GPTQ on $\hat{w} = \hat{X}^+ Xw$ instead of $w$.

This is what the Qronos (Zhang et al., 2026) algorithm does at its core, and it indeed improves the quantization quality, see the experimental results of Zhang et al. (2026).

**Theoretical Guarantees.** Theoretical guarantees about the output of Babai's algorithm directly carry over to GPTQ. First, there is an *absolute* guarantee on the error $\|Xw - Xv\|$ in terms of the lengths $L_{i,i}$ of the Gram-Schmidt vectors of the lattice:

**Theorem 3.1** (Babai (1986)). *The output $v$ of* BABAI *satisfies* $\|Xw - Xv\|^2 \leq \frac{1}{4}\sum_{i=1}^n L_{i,i}^2$.

Second, there is a *relative* error guarantee, relating the error to the minimally achievable error:

**Theorem 3.2** (Babai (1986)). *The output $v$ of* BABAI *satisfies* $\|Xw - Xv\| \leq \gamma \cdot \min_{v' \in \mathbb{Z}^n}\|Xw - Xv'\|$ *with*

$$\gamma \leq \sqrt{1 + \max_i \frac{1}{L_{i,i}^2}\sum_{j \geq i} L_{j,j}^2} \leq \sqrt{n+1} \cdot \max_{i \leq j} \frac{L_{j,j}}{L_{i,i}} \quad .$$

**Using Lattice Basis Reduction.** Theorem 3.2 suggests that the sequence $L_{1,1}, L_{2,2}, ...$ shouldn't ever increase much in order to obtain a good result. The classic way to make $L_{1,1}, L_{2,2}, ...$ not ever increase by much is by performing an LLL-like *lattice basis reduction*. This would give a guarantee on the $L_{i,i}$, and hence significantly improve the outcome of BABAI/GPTQ in theory (Nguyen & Vallée, 2010, Chapter 6, Theorem 3). A straightforward wrapper algorithm looks like this:

```
1: procedure WITHREDUCTION(X, w)
2:     (X_red, T) ← LATTICEBASISREDUCTION(X)
3:     [Here T ∈ ℤ^{n×n} is the base change matrix satisfying X_red = XT.]
4:     v_red ← BABAI(X_red, t = Xw)          ▷ Abuse of notation to pass t instead of w to BABAI.
5:     v ← Tv_red
6:     return v
7: end procedure
```

The intuition behind this algorithm is simple: What BABAI does is, given a lattice basis $X$ and a target vector $t$, it finds a lattice point close to $t$ and produces its (integer) coordinates $v$ with respect to the basis $X$. The "better" the basis $X$, the closer $Xv$ will be to $t$. Lattice basis reduction provides a "good" basis $X_{red}$ together with the base change matrix $T \in \mathbb{Z}^{n \times n}$ satisfying $X_{red} = XT$. Then we use BABAI on the reduced basis $X_{red}$, but with the same target vector $Xw$ as before, to compute a lattice point close to the target, which will be provided as coordinates $v_{red}$ with respect to $X_{red}$. Finally, we compute the coordinates of the point with respect to the original basis $X$ by applying $T$.

We note that if $X$ is not regularized enough, then $T$ and hence $v$ could potentially have large entries. This could be a problem when one needs to clip the values to a small quantization domain. Even without clipping, it could result in a bad accuracy of the final network, because large entries are a symptom of overfitting to the calibration data $X$.

We leave the experimental evaluation of WITHREDUCTION and related algorithms for future work.

ACKNOWLEDGMENT

We want to thank Rayan Saab for helpful conversations about this research.

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

## A    ALGORITHM DESCRIPTIONS

In this section we provide more detail on why our descriptions of BABAI and GPTQ indeed match the original algorithms.

For BABAI, compare with Nguyen & Vallée (2010, Chapter 6, Algorithm 1). To obtain our description, one needs to apply the following transformations to their algorithm:

- Our target (which they call $v$) is $Xw$.
- They use the notation $b_i^*$ to denote the $i^{\text{th}}$ Gram-Schmidt vector of the lattice basis. This corresponds to taking a QR-decomposition $X = QR$, and then letting $b_i^* = R_{i,i}Q_i$. Then:

$$\frac{\langle t, b_i^* \rangle}{\langle b_i^*, b_i^* \rangle} = \frac{\langle t, R_{i,i}Q_i \rangle}{\langle R_{i,i}Q_i, R_{i,i}Q_i \rangle} = \frac{\langle t, Q_i \rangle}{R_{i,i}} \tag{15}$$

So they round the same value as we do.

- Take a QL-decomposition instead of a QR-decomposition and process the loop in reverse order. (This accounts for reversing the basis.)

For GPTQ, take Frantar et al. (2023, Algorithm 1) and apply the following transformations:

- Remove the regularization factor $\lambda I$, since we assume it's already part of the lattice, as noted just before section 2.1.
- Note that what they call $X$ is $X^T$ in our paper.
- Instead of computing $L^{-1}$ as the Cholesky decomposition of $(X^T X)^{-1}$, compute a QL-decomposition of $X$ and invert $L$. This is equivalent, as explained in section 2.1.
- Choose block size $B = \infty$.

## B    EFFICIENT BABAI ALGORITHM

As explained at the end of section 2.2, in BABAI (and in GPTQ) one can replace $X \in \mathbb{R}^{k \times n}$ by $L \in \mathbb{R}^{n \times n}$, where $QL = X$ is the QL-decomposition of $X$, or in other words, $L^T L = X^T X$ is the reverse Cholesky decomposition of $X^T X$.

Plugging this into the BABAI algorithm yields the following algorithm, which is more efficient for memory and computation, because usually $k \gg n$, which helps in lines 3 and 6, and because there is no inner product computation in line 5.

1: **procedure** BABAI-CHOLESKY$(X, w)$
2:    Compute $QL = X$ or $L^T L = X^T X$. ▷ QL- or rev. Cholesky decomposition with $L_{i,i} > 0$
3:    $t^{(0)} \leftarrow Lw$
4:    **for** $i = 1, ..., n$ **do**
5:        $v_i \leftarrow \text{round}(\frac{t_i^{(i-1)}}{L_{i,i}})$
6:        $t^{(i)} \leftarrow t^{(i-1)} - v_i L_i$
7:    **end for**
8:    **return** $v$
9: **end procedure**

