# OpenReview forum: "The Lattice Geometry of Neural Network Quantization: A Short Equivalence Proof of GPTQ and Babai's Algorithm"
_ICLR.cc/2026/Conference — ICLR 2026 Poster_

### Official Review · Reviewer_Xcva · 2025-10-20

**Soundness:** 4
**Presentation:** 4
**Contribution:** 4
**Rating:** 8
**Confidence:** 4

**Summary:**

The authors show that a recently proposed algorithm for neural network quantization, GPTQ, is equivalent to Babai's well-established nearest-plane algorithm appearing in lattice problems.
Based on this equivalence and the long history of Babai's algorithm, they highlight that the algorithm is associated to known guarantees, and they describe possible improvements and extensions including a) the possibility of a pre-processing step involving basis reduction with the LLL algorithm, to improve performance guarantees; b) the extension to multilayer quantization.

**Strengths:**

Showing that a recent empirical approach to neural network quantization is in fact equivalent to a well-established tool from lattice geometry is important. Besides bringing clarification to the field, this also establishes connections that are potentially very fruitful: the paper notably mentions existing guarantees, and describes a number of convincing potential improvements. Some of these consequences also involve the analysis of Qronos, another recent quantization approach.  The paper is well-written, concise and clear.  The proof technique (proving equivalence to Babai-Proj-Rec, and to Babai_{\kappa_i} is very neat.

**Weaknesses:**

There is overlap between the contribution of this paper and a recent preprint by Chen et al. Chen et al provide numerical experiments, this paper does not.  Yet, the authors are explicit about this, both papers seem to have been produced at about the same time, and the absence of experiments in this submission actually also contributes to its elegance, concision, and focused viewpoint, which are all rather strengths.

It is not completely straightforward to check that the paper's description of GPTQ and Babai's algorithm fully matches the description of these algorithms in the corresponding references (by the way a chapter number would have helped when referring to Nguyen and Vallée).

**Questions:**

Can you detail why the algorithms you detail exactly match GPTQ and Babai's algorithm ?

Minor suggestions:
-elaborate on the impact of your viewpoint on the analysis of Qronos, which seems to be a distinctive feature of your analysis compared to that of Chen et al.
-include an explicit summary of the numerical improvements "mentioned in the paper" in section 3.
-write more explicitly that "what `t' is in the first recursion" (page 6) means "in the first call to line 6 of BABAI-PROJ-REC"
-clarify in section 2.3 what the "green sub lattice" is.
-to help the reader it may be useful to state after (3) that from now on k always assumed to be larger than n.
-top of page 3: the "projection" is a linear map, not a projection, right ?
-below (10): writing explicitly X_{>=2} (w+...) as an equation would seem clearer than spelling it out with words, and would ease the connection with (11);
-spell out GPTQ the first time this acronym is used

---

> ### Author Response · Authors · 2025-12-03
>
> We very much thank the reviewer for their feedback and their suggestions. In particular we want to thank for acknowledging that the absence of experiments contributes to the elegance and concision. Indeed the goal of the paper is to just provide an elegant short equivalence proof together with geometric intuition and context around the problem. We have addressed the suggestions and highlighted changes and additions in the paper in blue.
>
> **Description of GPTQ and Babai's algorithm in this paper vs original papers**
>
> We have added section A in the appendix that explains in detail how to get from the original algorithm descriptions to our descriptions, and we link to this section in the paper. We have also added a chapter and algorithm number for the citation(s) of Nguyen and Vallée.
>
> **Minor Suggestions**
>
> - Regarding Qronos. Indeed the concurrent work does not mention the handling of quantization over multiple layers, while we explain how to deal with this correctly. This in fact *precisely* recovers the Qronos algorithm. It yields better quantization quality as demonstrated experimentally by the Qronos paper. We have made this clearer in our paper.
> - We have made the part of the proof that refers to "t in the first recursion" clearer. It is now rephrased and mentions the line number as suggested.
> - At the same location, we have also replaced the prose mentioning "the product of X_{>=2} with (w+...)" with the explicit formula as you suggested.
> - Regarding mentioning that "from now on k always assumed to be larger than n", note that we do note just before section 2.1 that "from now on we will assume that X has linearly independent columns", which implies k >=n. (Just assuming k >= n wouldn't be enough.)
> - You asked "the 'projection' is a linear map, not a projection, right?". It is indeed a linear map, but it is also to be interpreted as a projection. We now call it a "linear projection" in the paper.

---

### Official Review · Reviewer_K3b5 · 2025-10-30

**Soundness:** 1
**Presentation:** 1
**Contribution:** 1
**Rating:** 0
**Confidence:** 1

**Summary:**

NA

**Strengths:**

NA

**Weaknesses:**

NA

**Questions:**

NA

---

### Official Review · Reviewer_w5cR · 2025-11-03

**Soundness:** 2
**Presentation:** 3
**Contribution:** 2
**Rating:** 4
**Confidence:** 3

**Summary:**

This paper establishes a formal connection between the problem of post-training weight quantization for linear layers in neural networks and the classical Closest Vector Problem (CVP) from lattice theory.

The central contribution of the work is a concise proof demonstrating that GPTQ is mathematically equivalent to Babai's nearest-plane algorithm. The authors discuss several consequences of this equivalence, including potential numerical stability improvements for GPTQ, handling quantization across multiple layers, and leveraging lattice basis reduction techniques to potentially enhance quantization performance.

**Strengths:**

I think the paper is easy to follow and has an interesting topic.

It takes expertise from both domains (Quantization and Lattice Geometry) to notice the equivalence between GPTQ and Babai. And it is valuable that Lattice Geometry knowledge from decades ago can help design better quantization algorithms.

The proof is more concise than the concurrent work mentioned in this manuscript.

**Weaknesses:**

The paper is purely theoretical and lacks experimental validation. This is an omission in a field as applied as model compression. The authors propose several compelling consequences in Section 3, most notably the use of lattice basis reduction (LBR) to improve quantization performance. However, these claims remain speculative. Critical questions regarding the computational overhead of LBR, its impact on the final model's generalization (the authors themselves note a risk of overfitting and generating large weight values ), and its actual impact on metrics like perplexity are left unaddressed.

The paper reads more like a well-written proof-of-concept than a complete ICLR-level publication. While the authors acknowledge an overlap in the manuscript with the concurrent work of Chen et al., it still impacts the novelty since the major conclusions of the two papers are the same.

**Questions:**

The paper acknowledges "significant overlap" with Chen et al. (2025). I do understand Chen et al is a concurrent arXiv paper, and it won't affect the review process of this manuscript. But I'm wondering, could the authors please elaborate on the specific, unique insights that your proof methodology provides?

---

> ### Author Response · Authors · 2025-12-03
>
> We thank the reviewers for their feedback, in particular for acknowledging that our paper is easy to follow and that the proof is more concise than the concurrent work, which is essentially our main goal. We address the concerns below. Changes in the paper are highlighted in blue.
>
> **Experimental Validation**
>
> You say that our paper lacks experimental validation, referring to "several compelling consequences in Section 3, most notably the use of lattice basis reduction (LBR) to improve quantization performance". Apart from the theoretical guarantees, we mentioned three other consequences in Section 3, and we will address the experimental validation of them one by one:
>
> - *Numerical Improvements.* Originally we had mentioned potential numerical improvements during our exposition. However, we have now removed those proposals and accordingly we also removed this paragraph from section 3. The reason is that for the previously mentioned numerical improvements to be possible, we would have to store the full matrix X (instead of just the Gram matrix X^T X), which is very memory intense if the number of calibration data points is large (k >> n), as common in practice. A better strategy would be to just stay with the Gram matrix, and use a higher precision for this matrix if necessary. We added a short note in section 2.1 that explains this, and notes that the QR decomposition is mostly useful for theoretical study, and in practice one would usually want to keep the equivalent Cholesky decomposition method.
> - *Correct Handling of Quantization Over Multiple Layers.* This indeed improves quantization quality. The main reason we don't include experiments is because our method *precisely* (!) recovers the Qronos algorithm, and the Qronos paper already provides experiments. We now made this clearer in the paper.
> - *Using Lattice Basis Reduction.* We hint at the possibility of using basis reduction and sketch a generic wrapper algorithm, however without specifying the precise basis reduction algorithm. We explicitly mention that we leave the experimental evaluation of such an algorithm to future work. We want to note that there are many (!) possible lattice basis reduction algorithms that are used in practice (usually variants of LLL), and they come with hyperparameters. For example, already the original LLL algorithm comes with two hyperparameters $\eta$ and $\delta$, variants of it usually come with more hyperparameters such as block size. We have experimented with basic lattice reduction algorithm and found that results are sensitive to these hyperparameters. Potentially it could help to regularize the basis reduction algorithm in a clever way. We also found that the runtime of the basis reduction algorithms heavily depend on the hyperparameters. Employing lattice basis reduction with the correct hyperparameters and regularization so that it works well with quantization and is computationally feasible will take time and would be a paper on its own, which is why we defer it to future work.
>
> In summary, we do not see the need for experimental validation. Our main contribution is a short, elegant proof of the equivalence of two existing algorithms. We do not propose any new algorithms that we would need to evaluate, except:
>
> - We explain how to correctly deal with quantization over multiple layers, but this precisely recovers Qronos, which has already been experimentally validated.
> - We hint on the possibility of using lattice basis reduction, but this is just a single paragraph in our paper and not the main point, and we explicitly mention concerns regarding overfitting/regularization and computational complexity and note that we leave the experimental valuation for future work.
>
> **Question regarding insights of proof methodology vs concurrent work**
>
> The main point, as you already noted, is that our proof is more concise and elegant. It fits on a single page, whereas the concurrent work uses several pages of calculations. The second point is that our proof highlights the core geometry of the problem. We mention in a previous section that the algorithms are related by a linear projection, and that whatever Babai does, if you project it down to R^n, it essentially yields what GPTQ does. Our proof shows this relation. We rewrite Babai in a recursive way, using the map $X_{\geq 2}$ and the weight update rule of GPTQ for updating the target $t$. By using this map (the embedding map $X$, see the diagrams), we essentially ignore everything that's killed by the projection anyway. We then demonstrate that this recovers the Babai algorithm, up to the correction factors $kappa_i Q_i$ (which don't affect the algorithm). So our proof exactly shows:
> - "embedding GPTQ using $X$ into $\mathbb{R}^k$ yields the Babai algorithm with some additional factors $\kappa_i Q_i$"
> - the factors $\kappa_i Q_i$ don't change the result of the algorithm
> Therefore the proof follows the geometry explained and plotted in figures in previous sections.

---

### Official Review · Reviewer_8GUU · 2025-11-04

**Soundness:** 2
**Presentation:** 2
**Contribution:** 2
**Rating:** 2
**Confidence:** 3

**Summary:**

In this work, the authors present a relatively straightforward equivalence proof between GPTQ (a well-known data-driven quantization approach for linear units in a neural network) and the classic Babai’s algorithm (a nearest-plane algorithm). After an in-depth discussion of both algorithms, including visualizations describing the underlying geometry, the equivalence proof is built by constructing recursive versions of both algorithms.

The paper concludes with a short discussion of the consequences of the equivalence, e.g., easier handling of quantization over multiple layers for GPTQ and the construction of an algorithm using lattice basis reduction.

**Strengths:**

The paper is well written and includes an in-depth discussion of background, making it easy to follow. The presented equivalence has potential for significant impact.

**Weaknesses:**

While the presented equivalence has potential for significant impact, the paper as it stands does not provide sufficiently strong evidence for this. The final consequences and future work section provides some interesting insights, but to warrant publication at ICLR, I believe that at least some of these observations have to be experimentally validated.

Most important in this regard is showing that the discussed numerical improvements hold and that the WithReduction algorithm provides the improvements the paper hints at. The correct handling of quantization over multiple layers would also be beneficial to study, though it may be hard to demonstrate.

**Questions:**

Not at this time.

---

> ### Author Response · Authors · 2025-12-03
>
> We thank the reviewer for their feedback, in particular for acknowledging that the paper is well written and easy to follow. We address the concerns below, and we highlighted changes in the paper with blue color.
>
> **Experimental Validation**
>
> You criticize the lack of "sufficiently strong evidence" for "some of these observations", asking for experimental validation. However, please note that this is primarily a theoretical paper, proving the equivalence of two existing algorithms, not proposing a new algorithm. We do however understand that section 3 might now have fully conveyed this point, so we made some adjustments. We will address all three "claims" from section 3 separetely below. (copied from other comment since the other reviewer asked about the same topic)
>
> - *Numerical Improvements.* Originally we had mentioned potential numerical improvements during our exposition. However, we have now removed those notes and accordingly we also removed this paragraph from section 3. The reason is that for the previously mentioned numerical improvements to be possible, we would have to store the full matrix X (instead of just the Gram matrix X^T X), which is very memory intense if the number of calibration data points is large (k >> n), as common in practice. A better strategy would be to just stay with the Gram matrix, and use a higher precision for this matrix if necessary. We added a short note in section 2.1 that explains this, and notes that the QR decomposition is mostly useful for theoretical study, and in practice one would usually want to keep the equivalent Cholesky decomposition method.
> - *Correct Handling of Quantization Over Multiple Layers.* This indeed improves quantization quality. The main reason we don't include experiments is because our method precisely (!) recovers the Qronos algorithm, and the Qronos paper already provides experiments. We now made this clearer in the paper.
> - *Using Lattice Basis Reduction.* We hint at the possibility of using basis reduction and sketch a generic wrapper algorithm, however without specifying the precise basis reduction algorithm. We explicitly mention that we leave the experimental evaluation of such an algorithm to future work. We want to note that there are many (!) possible lattice basis reduction algorithms that are used in practice (usually variants of LLL), and they come with hyperparameters. For example, already the original LLL algorithm comes with two hyperparameters $\delta$ and $\eta$, variants of it usually come with more hyperparameters such as block size. We have experimented with basic lattice reduction algorithm and found that results are sensitive to these hyperparameters. Potentially it could help to regularize the basis reduction algorithm in a clever way. We also found that the runtime of the basis reduction algorithms heavily depend on the hyperparameters. Employing lattice basis reduction with the correct hyperparameters and regularization so that it works well with quantization and is computationally feasible will take time and would be a paper on its own, which is why we defer it to future work.
>
> In summary, we do not see the need for experimental validation. Our main contribution is a short, elegant proof of the equivalence of two existing algorithms. We do not propose any new algorithms that we would need to evaluate, except:
> - We explain how to correctly deal with quantization over multiple layers, but this precisely recovers Qronos, which has already been experimentally validated.
> - We hint on the possibility of using lattice basis reduction, but this is just a single paragraph in our paper and not the main point, and we explicitly mention concerns regarding overfitting/regularization and computational complexity and note that we leave the experimental valuation for future work.

---

### Author Response · Authors · 2025-12-03

We thank all reviewers for their comments and feedback. We thank the area chair in advance for their work in this unusual and challenging situation due to the OpenReview bug. In light of the situation, we have submitted our responses at the end of the discussion period, since reviewers would not have been able to respond anyway, and formulated our response in a way that is hopefully helpful for the area chair to assess the points of discussion.

Two of the reviewers have raised concerns regarding experimental validation. We want to note that this is primarily a *theoretical* paper. We primarily prove the equivalence of two existing algorithms, and do not propose a new algorithm. Although in section 3 we suggest or hint on potential algorithmic improvements, this is only a minor part of our paper, and for each of the suggestions we have now either removed them from the paper (numerical improvements), explained how they were already experimentally validates in another paper (Qronos), or left for future work for specific reasons (basis reduction). Please see the response to the reviewers.

There was also one review which gave a score of 1 for everything and also a score of 1 for the confidence. We don't understand the meaning of that, we guess that the reviewer couldn't make a review. However, we found that their scores still made it into the averaging calculations. We ask the area chair to take this into account.

---

### Meta-Review · Area_Chair_VAX2 · 2026-01-02

**Summary:**

This paper presents a theoretical analysis showing the equivalence between the GPTQ algorithm - a data driven quantization approach for linear layers in a neural network and the Babai's algorihm for the closest Vector problem. The paper then discusses the consequences of this equivalence including potential numerical stability improvements for GPTQ, handling quantization across multiple layers, and leveraging lattice basis reduction techniques to potentially enhance quantization performance.

Based on the original reviews, reviewers made the following feedbacks:

Reviewer 8GUU finds the paper well-written with an in-depth discussion and easy to follow. The equivalence has a potential for significant impact. On the other hand, the reviewer mentions that the lack of experiments is an important weakness to support the results and in particular showing that the numerical improvement hold.

Reviewer w5cR argues on the positive side that the paper is easy to follow and has an interesting topic and it is also interesting that lattice theory can help to design better quantization algo. The proof is more concise than concurrent work.  On the other hand, the reviewer underlines the lack of experimental evaluation  which is seen as an omission in a field as applied as model compression. The computation overhead of lattice basis reduction is not discussed  neither its impact on model generalization. The paper looks like more a well-written proof-of-concept than a complete ICLR publication. There is also an overlap with Chen et al.'s paper.

Reviewer Xcva find t-on the positive side that this link with a well established tool in lattice theory is important: it establishes connections that are potentially very fruitful with potential improvements (such as the analysis of Qronos). The paper is well-written, concise and clear. The proof technique is very neat. On the other hand, the reviewer noted an overlap with Chen et al.'s paper. He noticed that there is not experiment but that is also in favor of having an elegant concise and focused viewpoint. He also mentioned that it is not straightforward to check if the description of the algorithm fully matches the description of the algorithms in the corresponding references.


 K3b5 did not write any review, its review and score are thus ignored.


Overall the paper presents a nice theoretical contribution showing the relationship between a known quantization algorithm and a famous result in lattice theory. All reviewers agreed on the fact the result is of interest with potential high significance. The paper is more concise and neat than other concurrent work.  The main weakness is the lack of experiments and maybe a form of paper which is not classic for ICLR.
Considering the lack of experiments, the paper could have been rejected, but considering the potential impact, I'm willing to discuss if we can champion it for acceptance.

**Reviewer Concerns:**

For reviewer 8GUU, authors answered that there is no need for an experimental evaluation since they showed the equivalence of two existing algorithms. They argue that they showed how to deal correctly with quantization over multiple layers and this precisely recovers the algorithm Qronos. For the lattice part, they leave experimental validation for future work.

For w5cR, they provided the same answer as the previous reviewer for the experiments. they also add a comment on their methodology vs concurrent work.

For reviewer Xcva, the authors answered how they get the descriptions
of the algorithm from the original version and added a section in the
Appendix on this topic.

**Reviewer Scores:**

Reviewer 8GUU gave a 2. It is difficult to assess if the reviewer would have increased his score, potentially yes but I am not sure it can go over 4.

w5cR gave a 4. It is also difficult to assess if the reviewer would have increased his score while one of his most important concern was the lack of experiments.

Xcva gave an 8, I think he would have kept his score.

---

### Decision · Program_Chairs · 2026-01-26

Accept (Poster)